# Home and Away: The Role of Non-Coding RNA in Intracellular and Intercellular DNA Damage Response

**DOI:** 10.3390/genes12101475

**Published:** 2021-09-23

**Authors:** Annabelle Shaw, Monika Gullerova

**Affiliations:** Sir William Dunn School of Pathology, University of Oxford, South Parks Road, Oxford OX1 3RE, UK; annabelle.shaw@lincoln.ox.ac.uk

**Keywords:** DNA damage, repair, RNA, exosomes, miRNA, lncRNA, bystander effect

## Abstract

Non-coding RNA (ncRNA) has recently emerged as a vital component of the DNA damage response (DDR), which was previously believed to be solely regulated by proteins. Many species of ncRNA can directly or indirectly influence DDR and enhance DNA repair, particularly in response to double-strand DNA breaks, which may hold therapeutic potential in the context of cancer. These include long non-coding RNA (lncRNA), microRNA, damage-induced lncRNA, DNA damage response small RNA, and DNA:RNA hybrid structures, which can be categorised as cis or trans based on the location of their synthesis relative to DNA damage sites. Mechanisms of RNA-dependent DDR include the recruitment or scaffolding of repair factors at DNA break sites, the regulation of repair factor expression, and the stabilisation of repair intermediates. DDR can also be communicated intercellularly via exosomes, leading to bystander responses in healthy neighbour cells to generate a population-wide response to damage. Many microRNA species have been directly implicated in the propagation of bystander DNA damage, autophagy, and radioresistance, which may prove significant for enhancing cancer treatment via radiotherapy. Here, we review recent developments centred around ncRNA and their contributions to intracellular and intercellular DDR mechanisms.

## 1. Introduction

Eukaryotic cells are frequently exposed to external and endogenous insults that cause DNA damage and threaten genomic stability [1]. A sensitive signalling cascade called the DNA damage response (DDR) therefore functions to sense and repair genetic lesions or to induce apoptosis in cells with irreparable damage, thus protecting genome integrity [2,3]. Double-strand DNA breaks (DSBs) represent one of the most cytotoxic forms of DNA damage, with a single unresolved DSB being capable of endangering overall cell health [3,4]. DSB repair is mediated by two major pathways: homologous recombination (HR) and non-homologous end-joining (NHEJ). Briefly, NHEJ is a fast, error-prone repair mechanism that does not require a homologous template. This means that it remains active throughout the cell cycle but is preferential during G1 [5,6]. NHEJ is initiated by the rapid binding of the Ku70-Ku80 heterodimers to the DNA ends at either side of the DSB, while end resection is limited by 53BP1. This triggers the recruitment of DNA-PKcs to form the DNA-PK holoenzyme, which tethers the ends together. The subsequent recruitment of downstream NHEJ factors, such as Artemis, XRCC4, DNA ligase 4 (LIG4), and XLF, then enable either the direct ligation of the blunt ends or end processing followed by ligation [7,8,9,10]. HR, on the other hand, is a high-fidelity mechanism that predominates in the S and G2 phases of the cell cycle, utilising the sister chromatid as a homologous template for repair. Initiated by the DSB sensor MRE11-RAD50-NBS1 (MRN) complex and its accessory factor CtIP, end resection is critical for commitment to HR. Short- and long-range end resection are facilitated by MRN, CtIP, BRCA1, and EXO1/DNA2 to generate single-stranded DNA (ssDNA) overhangs. These overhangs are protected by the immediate binding of RPA until BRCA2 mediates RPA displacement by RAD51 to form a nucleoprotein filament. The nucleoprotein filament then invades the template DNA duplex at a region of sequence homology, generating a D-loop intermediate and faithfully restoring the damaged strand without information loss [4,6,8,9,10].

In recent years, many studies have suggested critical roles for non-coding RNA (ncRNA) in DDR [11,12,13,14,15]. Within the cell, ncRNA species that contribute to DNA repair, particularly to that of DSBs, can be transcribed both proximally and distally to the site of DNA damage [1,5]. As both can influence activity at the break site, they have been designated as cis and trans mechanisms of RNA-dependent DDR, respectively. Nevertheless, RNA and its associated structures, such as R-loops, are also implicated in the induction of DNA damage, including DSBs, and genome instability [16]. Therefore, the tight regulation of RNA activities is required to maintain genome integrity and to resolve insults. RNA can additionally be exported into extracellular vesicles. Various species of messenger RNA (mRNA), microRNA (miRNA), and other ncRNA have been identified as encapsulated within a subset of extracellular vesicles that have a diameter of 40–100 nm, termed exosomes [17]. Following the discovery that exosome-mediated RNA transfer represents a key mechanism of intercellular communication and functional regulation [18], a growing body of evidence has implicated exosomal RNA in intercellular DDR mechanisms [19]. Intercellular DDR encompasses damage induction, genomic instability, and radioresistance within bystander cells, which have not been exposed to radiation themselves but are in the vicinity of directly irradiated cells [20]. Here, we review recent developments in the rapidly growing field of RNA-dependent DDR, focusing on the contributions of cis and trans ncRNA species to intracellular DNA repair mechanisms as well as the effects of specific intercellular RNA transfer on DDR in both donor and recipient cells. It should be noted that although there is some evidence for rRNA and tRNA involvement in DDR, these are not covered in this review.

## 2. Intracellular RNA-Dependent DDR Mechanisms

### 2.1. RNA Acting in Trans

#### 2.1.1. lncRNAs Contribute to DSB Repair

The activation of DNA repair by RNA transcripts synthesised in trans has been widely studied and reviewed [4,8,21,22]. These RNA transcripts are commonly ncRNA species, which can be further classified into long non-coding RNA (lncRNA) or small non-coding microRNA (miRNA) [4]. lncRNAs are over 200 nucleotides in length, transcribed by RNA polymerase II (RNAPII), and processed into mature RNA transcripts that can function in the nucleus or cytoplasm [8]. A growing body of evidence has implicated many lncRNA species in DDR and DNA repair mechanisms at various levels, including action as DSB sensors, transducers, or effectors [22]. Here, we discuss a selection of recently reported examples of lncRNA species with important functions in DDR, with emphasis on the DSB repair pathways.

NHEJ offers a rapid but low fidelity means of repairing a DSB via the binding of the Ku70-Ku80 heterodimer to the DNA ends at the break site, facilitating the recruitment of end-processing factors such as LIG4, which can resolve the DSB by means of the direct ligation of the DNA ends [2]. Many lncRNA species have recently been identified as enhancers of NHEJ efficiency, including LINP1 [23], LRIK [24], lnc-RI [25], and NIHCOLE [26], discussed below. These and other examples have been summarised in Table 1 to convey the diversity of lncRNA contributions to DDR.

LINP1 was first identified in triple-negative breast cancer (MDA-MB-231) cells as an interacting partner of Ku80 and DNA-PKcs by RNA pulldown and immunoprecipitation experiments [23], and the same methods confirmed this observation in cervical cancer (HeLa) cells [37]. It was initially proposed that LINP1 acted as a scaffold that was recruited to stabilise Ku80 and DNA-PKcs complexes at the repair site [23]. More recent work from Thapar et al. [27] aimed to investigate this interaction and its contribution to NHEJ. Indeed, their use of size exclusion chromatography small-angle X-ray scattering (SEC-SAXS) established that LINP1 promotes the association of multiple Ku heterodimers across the break to prolong the lifetime of the synaptic complex (Figure 1). Interestingly, LINP1 was found to act redundantly with the NHEJ accessory protein, PAXX. Single-molecule assays utilising molecular forceps to imitate DSBs showed LINP1 to be even more effective than PAXX at bridging DNA ends [27], thus highlighting the increasing importance of studying RNA molecules as key drivers of DNA repair. A similar scaffolding role was suggested for NIHCOLE, whereby its interaction with Ku80 promotes the formation of multimeric NHEJ complexes to increase ligation efficiency [26]. However, in contrast to the role of LINP1 in long-range synapsis [27], NIHCOLE is implicated in short-range synapsis [26]. While long-range synaptic complexes can provide the initial tethering of the DNA ends, it is the short-range synaptic complex that catalyses their precise alignment and ligation [38], distinguishing the functions of the two scaffolding lncRNAs. Furthermore, single-molecule atomic force microscopy revealed hyper-stoichiometric clustering of NIHCOLE and Ku80, such that around three molecules of NIHCOLE could recruit 14 Ku80 molecules, proposing a potential role for NIHCOLE in promoting phase separation as a means to drive NHEJ via “repair hubs” [26]. Another lncRNA, LRIK, was also found to interact with the Ku70-Ku80 heterodimer. However, its mechanism of NHEJ promotion differs from LINP1 and NIHCOLE. Chromatin immunoprecipitation (ChIP) in combination with AsiSi-induced DSBs in LRIK-knockdown cells revealed a dependence of Ku heterodimer accumulation at DSB sites, as well as the recruitment of downstream DNA-PKcs and XRCC4, on LRIK expression. LRIK was also found to directly bind damaged DNA, as shown by psoralen cross-linked chromatin isolation by RNA purification (ChIRP)-qPCR. In combination with the observations that LRIK loss downregulates γH2AX foci formation, these results implicate LRIK in the sensing of DSBs and the subsequent signalling to promote NHEJ, which is mediated by its interaction with Ku70 [24]. Finally, lnc-RI contributes to NHEJ by stabilising LIG4 expression, evidenced by reduced LIG4 at both the mRNA and protein level when lnc-RI was knocked down [25]. Prior work has also suggested a role for lnc-RI in HR via its ability to stabilise RAD51 mRNA [28], demonstrating that one lncRNA species can contribute to DNA repair via multiple mechanisms. Interestingly, both studies suggest that lnc-RI acts as a competitive endogenous RNA (ceRNA) for its respective target mRNAs [25,28]. ceRNAs have been described as non-coding RNA molecules that can be targeted by the same miRNA species as a specific coding RNA, providing additional regulatory potential via crosstalk between lncRNA and miRNA [39]. In this case, lnc-RI binds competitively via its 3′UTR to miR-4727-5p to enhance LIG4 expression [25] and to miR-193a-3p to regulate RAD51 expression [28]. Taken together, these results highlight the complexity of the non-coding RNA contribution to DNA repair mechanisms and demonstrate how their co-operation can lead to the fine-tuning of this process.

#### 2.1.2. miRNAs Can Modulate DNA Repair Mechanisms

miRNA represents another source of DNA repair modulation in trans. Transcribed by RNAPII at intragenic and intergenic regions, they undergo multi-stage processing by Drosha, DGCR8, and Dicer into mature miRNA duplexes with an average of 22 nucleotides [40,41]. The mature miRNA then associates with Argonaute (Ago) protein family members to generate the RNA-induced silencing complex (RISC). Following the degradation of the non-targeting miRNA strand, RISC mediates post-transcriptional gene silencing, termed RNA interference (RNAi), based on sequence complementarity between the miRNA guide strand and a target mRNA [42]. While perfect complementarity and Ago2 endonuclease activity lead to the degradation of target mRNA, sequence mismatches result in translational inhibition and are more common in animal cells [41].

Based on evidence that miRNA expression can both be regulated by DNA damage and impact its repair through RNAi [42], the contribution of miRNA to DDR is well-studied. It has been shown that certain DNA repair factors, BRCA1 and ATM, can enhance miRNA biogenesis and processing via interaction with Drosha or Drosha-associated proteins [43,44]. This suggests that DNA damage is responsible for the upregulation of a subset of miRNAs that may influence DDR and DNA repair mechanisms. Accordingly, many miRNA species that target DDR and HR factors were enriched following irradiation; however, many others were downregulated (summarised in [45]). Examples of miRNA species that can contribute to DNA repair modulation by means of the direct targeting of repair factors have been summarised in Table 2.

As seen in Table 2, some miRNAs can directly target more than one factor in a particular repair pathway, increasing their influence over that pathway. For example, miR-1255b, miR-148b*, and miR-193b* are all capable of controlling BRCA1, BRCA2, and RAD51 levels [11], thus providing multiple opportunities for miRNA-mediated control over HR. It has been proposed that the downregulation of HR factors by these particular miRNAs, shown in Figure 1, is responsible for NHEJ induction in the G1 phase of the cell cycle. As the sister chromatid is absent in G1, this serves to protect genome integrity and prevent loss of heterozygosity [11]. Some miRNA species, on the other hand, can exert their influence over multiple pathways. This includes miR-101, which can modulate NHEJ and HR by targeting DNA-PKcs or ATM, respectively [46]. As mentioned previously, miRNA can also work antagonistically with ceRNA [39] to provide an additional layer of control over DNA repair protein expression. Together, these findings demonstrate that miRNA directly contributes to the fine-tuning of the DDR and DNA repair mechanisms via RNAi. More recent work has additionally identified miRNA species acting indirectly to modulate DNA repair. Zhang et al. [47] identified miR-129-3p as an indirect modulator of NHEJ. Rather than directly downregulating the expression of a core DSB repair protein, miR-129-3p was found to silence SUMO-activating enzyme subunit 1 (SAE1) expression. This inhibited the sumoylation of XRCC4, preventing its nuclear localisation and subsequent DSB repair via NHEJ [47]. Similarly, Ge et al. [48] characterised miR-27a as an enhancer of HR via its downregulation of ZEB1, an ATM-interacting protein [48]. Overall, the studies reported here outline the diverse contributions of various miRNA species to DNA repair, both direct and indirect.

#### 2.1.3. Emerging Role of snRNA in DDR

Another class of small ncRNA, small nuclear RNA (snRNA), have recently emerged as novel players in DDR [60,61]. snRNAs are involved in pre-mRNA splicing, associating with proteins to form the small nuclear ribonucleoprotein (snRNP) particle subunits of the spliceosome [62]. Employing qRT-PCR, it was shown that U1 snRNA levels were reduced in response to UV irradiation. This reduction correlated with a transient increase in the alternative cleavage and polyadenylation isoforms of DDR-associated genes, thus reducing the number of full-length mRNA molecules [60]. Therefore, U1 snRNA plays an important role in the regulation of RNA processing and gene expression during DDR. Moreover, U2 snRNP complexes were subsequently shown to promote genome stability. This is achieved by a combination of two mechanisms. First, U2 snRNP maintains the transcription of essential repair proteins, including RAD51 and ATM, and aids in BRCA1 recruitment to damage sites, thus acting as a novel HR factor itself. The second mechanism acts to inhibit R-loop formation (discussed in Section 2.2.2), preventing R-loop-induced DNA damage and genome instability [61]. However, the contribution of U2 snRNA to these mechanisms is not indicated, meaning that further clarification would be useful to determine if these effects are more dependent on the RNA or protein content of the spliceosome. Nevertheless, interesting links between splicing and DDR clearly exist, which should be explored further in future studies.

Interestingly, another study reported a new class of small ncRNA that were associated with single strand DNA damage repair (sdRNA). These molecules appear to be BRCA1/RNAi multi-protein complex dependent and to activate the PalB2/Rad52 complex, in order to promote repair at R-loop rich transcription termination sites. It is not clear whether these sdRNAs are actually derived from R-loops or their vicinity, or if they act in cis or trans. However, this mechanism serves to prevent the genome instability that is associated with R-loop formation [63].

### 2.2. RNA Acting in Cis

#### 2.2.1. Damage-Induced Transcripts Can Modulate DNA DSB Repair

It is well reported that transcription can be transiently activated at DSB sites, contributing to efficient DNA repair [64]. A key development in the field was the discovery of de novo RNA transcripts generated in close proximity to break sites, which appears to be specific to DSBs and conserved between yeast, plants, insects, and mammalian cells [12,13,14,65]. These small ncRNAs were designated as DSB-induced RNAs (diRNAs) and DNA damage response small RNAs (DDRNAs) [12,13]. DDRNAs are characterised as being 20-35 nucleotides in length and as sharing sequence homology with the damaged locus [13]. They have been shown to be processed in a Dicer-dependent manner from RNAPII-transcribed precursors, which are called damage-induced long non-coding RNAs (dilncRNAs) [13,15]. This is supported by reports of phosphorylated Dicer (p-Dicer), which accumulates in damaged nuclei and promotes the turnover of nuclear damage-induced RNA [66]. dilncRNA transcription has been attributed to the canonical activation of RNAPII via phosphorylation at Ser-2 or Ser-5 of its C-terminal domain (CTD) [15] and its recruitment to DNA ends by the pre-initiation complex, generating a functional promoter at the break site [67]. However, it was recently reported that the tyrosine kinase c-Abl could phosphorylate Tyr-1 (Y1P) of the RNAPII CTD and catalyse its activation at promoter-associated DSBs. RNAPII Y1P was proposed to generate damage-responsive transcripts (DARTs), which could then be converted to dsRNA via a mechanism involving DNA–RNA hybrid intermediates serving as antisense promoters [68]. Similar to dilncRNA, these dsRNA could subsequently be processed by p-Dicer with consequences for DDR signalling and DNA repair, which will be discussed below [15,68].

diRNAs are similar small ncRNA species of 21-24 nucleotides in length, which have largely been identified and characterised in *Arabidopsis thaliana* but have also been observed in mammalian cells. Also dependent on Dicer or Dicer-like proteins for their biogenesis [12], they exert their influence on DNA repair via Ago2 effector molecules [12,69,70]. Briefly, diRNA-associated Ago2 facilitates the recruitment of RAD51 [69] as well as the chromatin remodelling enzymes MMSET and Tip60 [71] to DSB sites. The resulting chromatin relaxation allows enhanced access of RAD51 and BRCA1 to the damage site, thus facilitating repair via HR [71]. This activity is an example of the in cis regulation of DSB repair by a small RNA species. Curiously, there appear to be discrepancies in the literature such that it is unclear whether diRNA in mammalian cells are distinct from DDRNA, as some studies seemingly combine observations and discuss the two interchangeably. To avoid confusion, we will continue to examine DDRNA only.

DDRNAs were identified based on their ability to restore the formation of specific DDR foci that had been abrogated by RNase A treatment, and their action was found to be dependent on the MRN complex [13]. Therefore, DDRNAs clearly play an important role in the activation of repair at damage sites. To investigate whether this was a primary role in lesion recognition or whether they were involved in secondary recruitment of DDR factors, Francia et al. [72] employed laser micro-irradiation in combination with Dicer/Drosha knockdown. While the recruitment of the primary repair factor and MRN complex component, NBS1, to DNA damage sites was unaffected by the loss of Dicer or Drosha, that of secondary MDC1 and 53BP1 were significantly reduced. Alongside evidence that RNase A-sensitive MDC1 and 53BP1 foci can be restored by synthetic DDRNAs in a sequence specific-manner, this demonstrates that DDRNA are dispensable for primary recognition and DDR activation but that DDRNA is required for secondary repair factor recruitment and signal amplification [72]. This conclusion is strengthened by findings that the MRN complex is necessary and sufficient for active RNAPII localisation to DSBs and subsequent dilncRNAs synthesis [15,73], thus suggesting that DDRNAs function downstream of MRN. It was further shown that 53BP1 foci could be restored by the specific localisation of fluorescently labelled DDRNA to the break site, suggesting that its function in DDR activation is mediated from the break itself [15]. Interestingly, RNA pull-down assays identified this sequence-specific localisation as mediated by dilncRNA–DDRNA interactions, demonstrating that dilncRNA functions as more than a DDRNA precursor [15]. Both species were then shown by RNA immunoprecipitation to bind 53BP1 via its Tudor domain [15], suggesting that DDRNA–dilncRNA complexes are responsible for 53BP1 recruitment and accumulation at DSB sites, and for DDR foci formation (Figure 1). Accordingly, the disruption of DDRNA–dilncRNA interaction by antisense oligonucleotides significantly reduced 53BP1 localisation to DSB sites, while γH2AX was unaffected [15]. dilncRNA was further implicated in driving the liquid–liquid phase separation of 53BP1, leading to foci formation [67]. DDRNA and dilncRNAs therefore represent RNA species that are both generated from and act at damage sites to mediate DDR, exemplifying RNA-dependent DDR in cis. Interestingly, subsequent work from Gioia et al. [74] recapitulated the influence of DDRNA on 53BP1 action and NHEJ, adding the novel finding that it can be modulated by the pharmacological agent, enoxacin, to enhance DNA repair and to promote the survival of damaged cells [74]. Enoxacin is an antibiotic that has been shown to enhance Dicer-dependent miRNA processing and that has been proposed as an anti-cancer drug [75], highlighting the interplay between cis and trans ncRNA species. Overall, the work discussed here supports direct roles for damage-induced transcripts in DDR. In particular, dilncRNA and DDRNA appear critical for DSB repair.

Finally, many reports have examined a novel role for antisense transcript RNA as a template for DSB repair via HR in yeast cells [76,77,78]. This was reportedly driven by Rad52-mediated strand exchange [77] and translesion DNA polymerase ζ activity [78], preferentially occurring in cis [76]. Similarly, nascent pre-mRNA was identified in complex with classical NHEJ components upon DSB induction in human cells. Therefore, a mechanism of RNA-templated repair was proposed in which nascent transcripts generated from transcriptionally active DSB sites could serve as templates for error-free NHEJ [79]. While both mechanisms occur in cis and involve DNA:RNA hybrid intermediates, they are distinct from one another. This was made clear by Meers et al. [78], who showed Rad52-driven RNA-templated repair to be enhanced upon the loss of NHEJ components. Although further investigation is necessary to determine whether these mechanisms can act with complementarity in human cells, it is clear that the nascent RNA generated from transcriptionally active damage sites can play an important role in maintaining sequence fidelity during DSB repair. Therefore, endogenous transcripts may hold equal relevance to de novo transcripts during RNA-dependent DDR in cis.

#### 2.2.2. DNA:RNA Hybrids/R-Loops Contribute to HR

dilncRNA pairing with single stranded DNA after resection has been shown to facilitate DNA:RNA hybrid formation [80], which is consistent with earlier suggestions that RNAPII-generated transcripts would hybridise with template DNA [14]. DNA:RNA hybrids and the associated R-loops, which are formed by the displacement of one DNA strand to generate a triple-stranded structure [81], have been well-studied in recent years. Many studies have detailed their positive contributions to DNA repair [14,80,82,83,84,85], meanwhile others focus on their detrimental effects, including their potential to induce genome instability and how their resolution is critical for DNA repair [86,87,88,89,90]. For example, it was recently shown that the DEAD box RNA helicase DDX5 is a critical regulator of R-loop resolution and its depletion induced HR defects [90]. This activity was enhanced by BRCA2 [91] and implies that the BRCA2/DDX5-dependent removal of R-loops is critical for allowing HR to proceed.

Here, we will focus on the recently reported contribution of DNA:RNA hybrid presence to DSB repair via HR. It was previously shown that DNA:RNA hybrids accumulate preferentially in the S/G2 phase of the cell cycle and downstream of DNA end-resection, hence pointing towards a role in HR [80]. Accordingly, BRCA1 was shown to bind DNA:RNA hybrids with a similar affinity to dsDNA, and BRCA1 foci formation could be downregulated by RNase H1 overexpression [80]. Recent work from Ouyang et al. [84] recapitulated the enhancement of HR stimulation by local transcription, using a tetracycline-inducible direct repeat GFP reporter assay. They then extended this assay to include a fusion RNA species that could be tethered to I-SceI-induced DSBs by dCas9 targeting, which led to the finding that RNA tethering to a site 5′ of the DSB could promote HR in the absence of transcriptional activity. Due to this effect being dependent on the length, sequence, and orientation of the RNA transcript, they concluded that the stimulation of HR relies on DNA:RNA hybrid formation at DSB sites [84]. DNA–RNA immunoprecipitation (DRIP)-ddPCR was also employed to demonstrate the important role of the RAD51-binding protein, RAD51AP1, in generating DNA:RNA hybrids both at the DSB and in donor DNA. This led to the discovery of DR-loops, a novel structure containing both DNA:DNA and DNA:RNA hybrids. It was thus proposed that RAD51AP1-dependent R-loop formation aided in RAD51-dependent D-loop generation [84], contributing to HR because D-loops represent bona fide HR intermediates that facilitate template-based repair [92].

Novel findings recently reported by Liu et al. [93] support the accumulating evidence that transient DNA:RNA hybrids represent an essential intermediate for DNA repair via HR. Their model proposes that DNA:RNA hybrids function to promote the displacement of the 5′-strand and to ultimately protect the exposed 3′-ssDNA overhangs during end resection [93]. They demonstrated that the CRISPR-Cas9-mediated knockout of CtIP inhibited DNA:RNA hybrid fluorescence signals at laser micro-irradiation sites, as did MRE11 inhibition using mirin [93]. This is consistent with earlier studies that employed DRIP-qPCR to establish that the knockdown of EXO1 and CtIP impedes DNA:RNA hybrid formation [80]. Taken together, these results show that DNA:RNA hybrid formation occurs after the initiation of end resection by MRN-CtIP. Nevertheless, Liu et al. [93] sparked controversy by proposing RNA polymerase III (RNAPIII) as the primary mediator of transcription at break sites and DNA:RNA hybrid formation, as opposed to the widely accepted RNAPII [13,15,67,68,70,73,94]. Evidence provided by mass spectrometry, laser micro-irradiation, and ChIP-qPCR appears to show that RNAPIII is the only RNA polymerase recruited to DSBs, and that transcription of the RNA participating in DNA:RNA hybrid formation is catalysed by RNAPIII [93]. This calls into question the existing model of dilncRNA synthesis and RNAPII-dependent DNA repair that has been discussed above, suggesting that further work is required to investigate these opposing models of transcription-related DNA repair. Despite the evidence that DDRNAs are produced at the break by RNAPII [13], Liu et al. [93] attempt to assimilate their presence at DSBs into their model by suggesting that they result from the degradation of the RNA component of the hybrid, which then allows access of RAD51 to the ssDNA to complete end resection [93]. Interestingly, both RNAPII and RNAPIII are reportedly recruited to DSBs in an MRN-dependent manner [15,73,93]. Therefore, it may be possible that the two models are not mutually exclusive and that they represent two complementary mechanisms that can both be utilised by the cell. This is endorsed by the previously mentioned finding that it is the RNA molecule itself and its presence at the break rather than transcriptional activity that contributes to HR promotion [84]; therefore, it should not matter which polymerase catalyses the transcription. Nonetheless, the observation that RNAPIII subunit knockdown impairs RAD51 and RPA32 foci formation [93] corresponds with the conclusion of the aforementioned study: DNA:RNA hybrids function upstream of RAD51 and may contribute to repair via D-loops [84]. Overall, there have been interesting developments in the study of DNA:RNA hybrid contribution to DSB repair via HR, but further work is needed to consolidate the exact mechanism.

The source of RNA engaging in hybrid formation is also still under debate. Using DRIP-qPCR to assess the formation of DNA:RNA hybrids induced by I-PpoI nuclease cleavage in both intragenic and intergenic regions, D’Alessandro et al. [80] propose that dilncRNAs are responsible for hybrid formation. Alongside further genome-wide analysis of AsiSi-induced DSBs [86], they showed DNA:RNA hybrid formation to be independent of the transcriptional status of the location prior to DSB induction [80]. However, this was disputed by a more recent report from Bader et al. [95]. Using the same previously published AsiSi DRIP-seq dataset [86], they re-performed the analyses with sites grouped according to their transcriptional efficiency. This was to address reports that intergenic sites can contain AsiSi recognition sites with high transcriptional activity, meaning that “intergenic” does not necessarily equate to low transcriptional activity [95]. Interestingly, their analyses showed a significant enrichment of R-loops at highly transcribed loci, implying a dependence of R-loop formation on pre-existing transcription at the damage site. They therefore conclude that the RNA participating in hybrids is transcribed prior to damage induction, as opposed to being damage-induced transcripts [95]. This supports the proposed mechanisms of RNA-templated DNA repair (discussed in Section 2.2.1), which utilise nascent RNA transcripts in cis and generate DNA:RNA hybrids. Another relevant study used single strand annealing assays to demonstrate in *Saccharomyces cerevisiae* that hybrids generated from mRNA in cis, but not in trans, had the potential to influence HR by acting as a source of recombinogenic damage to promote further repair [89]. While this does not necessarily distinguish the possibilities of de novo or pre-existing transcription providing the R-loop-participating RNA, it does highlight DNA:RNA hybrid/R-loop formation and its possible influence on mechanisms of DNA repair as another example of RNA-dependent DDR regulation occurring in cis. This contrasts with previous suggestions that homologous transcripts were synthesised in trans and then complex with RAD52 to facilitate a sequence-specific search for homologous DSB target sites. However, it was noted that a cis mechanism would enhance the efficiency of repair via hybrid formation in this experimental setup [96]. Whether the RNA transcripts contributing to DNA:RNA hybrids are the result of de novo or pre-existing transcription remains to be seen. Nonetheless, the existing evidence appears to support hybrid formation as another vital contribution to DNA repair in cis, whose presence must be tightly regulated to ensure that genome stability is maintained.

Overall, the dependence of intracellular DDR mechanisms on RNA transcripts produced both in cis and trans has become increasingly clear. Many species of non-coding RNA, including various lncRNAs and miRNAs, dilncRNA, and DDRNA/diRNA, directly contribute to the repair of DNA lesions. Their mechanisms are diverse and encompass scaffolding functions, the modulation of repair factor expression, and secondary structure formation. Key examples have been summarised in Figure 1. Nevertheless, further studies are still required to consolidate these mechanisms and to address the controversial findings that have been recently described. In particular, it remains to be seen whether a unified model for damage-induced transcription and subsequent DNA:RNA hybrid formation can be elucidated, which would prove influential to the field of RNA-dependent DDR.

## 3. Intercellular DDR Mechanisms

### 3.1. The Radiation-Induced Bystander Effect

In addition to the intracellular mechanisms of RNA-dependent DDR, accumulating evidence has suggested that RNA plays vital roles in intercellular communication and subsequent DDR. Exposure to ionising radiation (IR) poses a threat to genome integrity by inducing various forms of DNA damage, including single-strand breaks (SSBs), DSBs, and DNA crosslinks. Cells therefore utilise DDR in response to IR, inducing cell cycle arrest and allowing DNA repair, e.g., via HR or NHEJ in the case of highly cytotoxic DSBs [97,98]. The radiation-induced bystander effect (RIBE) is an interesting, if not somewhat perplexing, phenomenon in which the neighbouring cells of those targeted by radiation can exhibit signs of radiation-induced damage themselves [99]. Despite not being directly traversed by radiation, these so-called bystander cells can display signs of DNA damage, genomic instability, apoptosis, or decreased survival, termed RIBE endpoints [100] (Figure 2). First reported in 1992 [101], RIBE has been shown to occur over a remarkably long distance, with apoptosis and micronuclei induction observed in bystander cells up to 1 mm (approximately 50–75 cell diameters) away from the irradiation site in a 3D human tissue model [99]. The underlying mechanisms remain elusive while being widely studied; however, it is evident that exosomes play an important role in transmitting RIBE-inducing signals [19,102,103,104,105]. It also appears to be a combination of the RNA and protein content of exosomes contributing to this effect [102], which will be discussed henceforth. Interestingly, RIBE has also been described as a long-term phenomenon, with bystander damage even observed in cells treated with exosomes from the progeny of both irradiated and bystander cells [102]. Therefore, its influence is wide-reaching and warrants further study.

#### 3.1.1. Exosomes and Their Composition Are Influenced by Ionising Radiation

Many studies have reported alterations in exosome cargo upon the irradiation of the cells from which they are secreted [106,107,108,109,110]. For example, mass spectrometric analysis of exosomes from head and neck carcinoma (FaDu) cells revealed 236 and 69 proteins that were up and downregulated upon 2 Gy (IR) treatment, respectively [107]. Among the proteins whose presence in exosomes were affected by IR, the most enriched biological processes from gene ontology (GO) analysis included RNA metabolism, gene expression, cell division regulation, and cell signalling [107]. The authors suggest that this can be attributed to the removal of these factors from cells no longer requiring these processes due to IR-induced cell cycle arrest [107]. While this is a valid point, it does overlook evidence that exosomes also act as vehicles for intercellular communication and that they can influence the activities of recipient cells [111]. Further studies in head and neck cancer cells also identified associations between IR-upregulated proteins and processes of migration [109], DNA repair, and reactive oxygen species (ROS) metabolism [108]. Due to the well documented connection between exosomes and RIBE, Abramowicz et al. [108] also searched their data for GO terms that could be hypothetically linked to RIBE, such as regulators of stress response or apoptosis. Despite IR-modulated proteins not being generally overrepresented in these processes, many proteins with roles in stress-induced transcription, MAPK cascade, apoptosis, or autophagy were identified as upregulated in exosomes upon IR treatment [108]. Therefore, this supports the notion that the protein content of exosomes may be a contributing factor to RIBE.

Other studies have focused on the RNA content of exosomes upon IR-treatment, particularly miRNA [109,110]. Interestingly, the exosomal presence of miRNA species associated with apoptosis and cell proliferation [109], as well as DDR [110], were also regulated in response to IR. Similarly, exosomes from cells that were irradiated with non-ionising ultraviolet B exhibited the upregulation of miRNA whose target genes are involved in autophagy, DNA damage, and DNA repair [112]. A comparison of IR-treated parental cells and their released exosomes also showed differences in miRNA content, including miR-3168, whose expression was increased in exosomes but downregulated in cells [110]. This is significant as miR-3168 targets include members of the ErbB signalling pathway, meaning that its upregulation in exosomes may lead to apoptosis in non-targeted recipient cells in patients treated with cisplatin [113]. Additionally, it suggests a mechanism for the active sorting of miRNAs into exosomes upon damage induction [110]. This is supported by findings that the RNA-binding protein YBX1 is involved in the loading of miRNA, such as miR-122, into exosomes [114], as well as in the specific packaging of many other abundant small ncRNA species, including tRNA, Y RNA, and Vault RNA [115]. Although this was not investigated under damage conditions, it is still relevant to the study of RIBE, as it implies active cellular control over the RNA composition of exosomes, which is likely to undergo changes upon alterations to cellular conditions. Taken together, these studies support a combinatorial role of proteins and miRNA in transmitting RIBE-inducing signals via exosomes, possibly including apoptotic mediators.

#### 3.1.2. Roles of Exosomal Protein and RNA in RIBE Induction

While experiments involving RNase treatment and the denaturation of exosomal proteins have concluded that RNA and protein work synergistically to transmit RIBE to bystander cells via exosomes [102], a dependence of this phenomenon on named exosome-packaged proteins has not yet been reported. That is not to say that proteins are not involved in RIBE; one example is soluble TNFα in conditioned medium, which was shown to mediate high-dose RIBEs in lung adenocarcinoma cells [116]. However, for the purposes of this review, we focus on recent advances that mainly implicate exosomal contents in this process. Most of these studies identify key roles for RNA species, particularly miRNA, in mediating RIBE. Examples are summarised in Table 3.

MicroRNA-21 (miR-21) is a well-reported example of a specific miRNA species functioning as a RIBE mediator [121,122,129,130]. Its increased expression in bystander cells was confirmed to be the result of uptake from the extracellular medium, evidenced by no change in pre-miR-21 levels in bystander cells [129], thus advocating intercellular communication as the key to RIBE transmission. Although it is not explicitly stated that this uptake occurs via exosomes, it would be a fair assumption based on independent evidence that many RNA species are enclosed within exosomes [115]. Decreased clonogenic survival, increased 53BP1 foci formation, and increased micronuclei induction are well-reported RIBE endpoints. These effects were observed upon the transfection of unirradiated MRC-5 fibroblast cells with a miR-21 mimic [129], thus simulating RIBE and demonstrating its dependence on miR-21. Similar results were observed in non-small cell lung carcinoma cells treated with a miR-21 mimic, showing that miR-21 upregulation is involved in increased ROS and associated DNA damage induction [121]. Interestingly, this study also observed a reduction in cell proliferation upon miR-21 inhibition [121]. They conclude that miR-21 upregulation in bystander cells, likely through endocytic uptake of exosomes, occurs rapidly to cause immediate DNA damage and then becomes downregulated at a later time point to inhibit proliferation [121]. The regulation of ROS can be attributed to miR-21-mediated control of SOD2 expression [122] and the TGF-β1 pathway [121], with the latter also being involved in micronuclei induction and the bidirectional control of miR-21 expression [130]. The TGF-β1 pathway has also been identified as the target of another miRNA in the context of RIBE: miRNA-663 (miR-663) [119]. While miR-21 is an activator of RIBE, miR-663 acts as a negative regulator to limit RIBE by suppressing TGF-β1 expression in a negative feedback loop, thus contributing to the distance-dependence of RIBE [119]. Taken together with Table 3, these studies exemplify the complex and critical roles of specific miRNA species in transmitting and controlling RIBE.

Interestingly, these studies contradict prior work from Dickey et al. [131]. Using medium transfer and co-culture methods in Dicer knockdown cell lines, they still observed bystander effects in the form of γH2AX foci when the total mature miRNA was depleted. They therefore concluded that miRNAs were not critical mediators of RIBE [131], which is in direct contrast to the more recent studies discussed above. However, it was later observed that a small number of canonical 5p miRNAs can still be synthesised in Dicer knockout cells, instead utilising direct Ago2 loading for their maturation [132]. As a result, this work does not necessarily rule out miRNA as direct contributors to RIBE. Overall, further investigation is still needed to unravel the specific contributions of miRNA and other species of ncRNA to RIBE. Understanding their influence over this phenomenon could have important implications for radiotherapy treatment, for example, in controlling the off-target effects of cancer treatment on healthy tissues.

### 3.2. Radiation-Induced Rescue Effect

Although bystander signalling in the context of RIBE appears to have largely negative impacts on recipient cells (e.g., DNA damage and apoptosis), it has also been shown to generate positive outcomes: radiation-induced rescue effects (RIRE) [133] and radioprotective effects [134] (Figure 2). RIRE is a phenomenon related to RIBE, in which feedback signals from non-irradiated bystander cells or their culture medium can limit the negative effects of irradiation and can “rescue” target cells [135]. This has been observed in both normal and human cancer cell lines as significant decreases in 53BP1 [133] and γH2AX [136] foci, micronuclei induction, and apoptosis [133] when irradiated and bystander cells were co-cultured. As these effects were statistically significant within 24 h [133,136], it implies that intercellular signalling may be responsible for enhanced DSB repair and reduced levels of apoptosis upon co-culture. Indeed, Lam et al. [137] identified the nuclear factor κB (NF-κB) as a major contributor to RIRE; they recapitulated the RIRE phenotype via 53BP1 foci measurement in HeLa cells, finding it to be abrogated by the NF-κB activation inhibitor, BAY-11-7082 [137]. The same group later confirmed the increased nuclear expression of active, phosphorylated NF-κB in irradiated HeLa cells when treated with the conditioned medium of bystander cells [138]. Together, these findings propose the activation of the NF-κB pathway in target cells as a key driver of RIRE.

A possible explanation for this increase in activity may be the reported downregulation of NF-κB-associated miR-34a-5p and miR-146a-5p in radiation-derived exosomes compared to unirradiated (Table 3) [110], leading to a slight reduction in NF-κB inhibition. However, recent findings have suggested that NF-κB activation is due to interleukin-6 (IL-6) secretion from bystander cells, resulting from the induction of autophagy by RIBE signalling [139]. The authors compare this to metabolic cooperation, a process in which cancer cell survival is supported by nutrients provided by normal cells [139]. This links back to Section 3.1.1, in which we mentioned the protein and miRNA species associated with autophagy being differentially regulated in radiation-derived exosomes. In particular, exosomal miR-7-5p was identified as a critical inducer of autophagy in bystander bronchial epithelial (BEP2D) cells [117]. This highlights the complex interplay between irradiation, exosomal contents, and the induction of both RIBE and RIRE. Interestingly, it was further shown that while the induction of autophagy in bystander cells can enhance RIRE, it is also able to reduce the damaging effects of RIBE [139]. This suggests a disconnect between RIBE and RIRE, questioning the idea that RIBE-induced damage is necessary to propagate RIRE. In addition to IL-6, a positive feedback loop involving the DDR factor, poly(ADP-ribose) polymerase 1 (PARP1), has been proposed to regulate NF-κB activation [140]. Media transfer experiments and the use of the PARP1 inhibitor, Olaparib, confirmed PARP1 involvement in RIRE in various carcinoma cell lines (HeLa, MCF7, CNE-2 and HCT116) [140]. Due to previously reported links between PARP1 activity and NF-κB-mediated transcription [141], the authors then investigated the reciprocal effects of PARP1 and NF-κB inhibition. They found NF-κB expression to be both transcriptionally and translationally reduced by PARP1 inhibition, and vice versa, advocating a positive feedback loop between them with the potential to influence RIRE induction [140]. This finding offers an attractive opportunity for cancer therapy. As RIRE can occur between bystander healthy cells and irradiated cancer cells [136,142], it has the potential to reduce the effectiveness of radiotherapy. Therefore, by employing PARP inhibitors in combination with radiotherapy, RIRE could be mitigated to enhance the radiosensitivity of cancer cells and to improve therapeutic outcomes. In summary, RIRE provide justification for the apparent negative effects of RIBE, including autophagy. This phenomenon also highlights the interplay between miRNA, DDR, and the immune response in controlling a single pathway to influence radiation response.

### 3.3. Radioprotective Effect

A similar but distinct radioprotective response has also been reported, which refers to the apparent reduction in damage induction upon the subsequent irradiation of bystander cells [134,143]. Although this is sometimes referred to as a radioadaptive bystander response, the term “radioprotective” will be used for the purposes of this review. This is to avoid confusion with the separate radioadaptive response, which refers to the reduced radiosensitivity that is observed following a low dose of priming radiation being administered to target cells (reviewed in ref. [144]). Early work has alluded to this protective effect by interrogating the proteome of bystander rainbow trout gills [145]. Although radioprotection was only hypothesised in this study, it was later observed in vivo within naive zebrafish embryos who were challenged with α particles after sharing media with irradiated embryos [146]. This was then recapitulated in vitro by Pereira et al. [147], who reported a reduction in γH2AX foci, thus radioprotection, in bystander embryonic zebrafish (ZF4) cells that were themselves subjected to low doses of γ irradiation [147]. Despite efforts to identify a secreted protein factor that may be responsible for these effects, the authors were unsuccessful and concluded that due to the short duration of their experiment, a factor other than protein may be involved in this so-called “early” bystander effect [147].

Many recent studies have used human cancer cell lines to investigate the role of exosomes in the radioprotective response [128,148]. Using head and neck cancer (BHY and FaDu) cell lines, Mutschelknaus et al. [148] demonstrated that the uptake of exosomes, regardless of whether the donor cells were irradiated or not, increased recipient cell proliferation [148]. On the other hand, Mrowczynski et al. [128] determined that the exosomes from donor cells treated with 3 Gy or 12 Gy radiation induced significantly higher rates of proliferation in recipient nervous system cancer cells (U87 glioblastoma cells, STS26T malignant peripheral nerve sheath tumour cells and SH-SY5Y neuroblastoma cells), compared to exosomes from unirradiated donor cells. They suggest that this increase in proliferation allows for improved survival upon subsequent exposure of these recipient cells to radiation [128]. Indeed, both studies reported significant radioresistance in the recipient cells of radiation-derived exosomes, which were measured via clonogenic survival assay [148] or apoptosis assay [128]. Mutschelknaus et al. [148] then investigated whether this could be attributed to enhanced DNA repair mechanisms. While they observed no effect on 53BP1 foci formation within one hour of irradiation and exosome treatment, there were significant reductions in 53BP1 foci within 6 and 12 h of irradiation and radiation-derived exosome treatment, compared to control exosome treatment [148]. Consequently, one mechanism underlying bystander radiation resistance may be the enhancement of DNA repair, albeit in a slightly delayed manner. Furthermore, the high-concentration RNase A treatment of exosomes abrogated their ability to impact DNA repair [148], thus suggesting an important role for exosomal RNA in communicating this phenotype. This is supported by previously mentioned findings that the miRNA content of exosomes is modulated by exposure to IR [109,110]. As shown in Table 3, IR-induced dysregulation of specific miRNA species can enhance the radioresistance of recipient cells. In particular, miR-365 was found to be downregulated in radiation-derived exosomes from both FaDu [110] and U87 cells [128]. As this miRNA usually inhibits the expression of Bcl-2, Cyclin D1, and PI3K, its downregulation leads to increases in recipient cell proliferation that could help increase their survival following radiation treatment [128]. Another example is miR-889, whose upregulation in radiation-derived exosomes leads to reduced DAB2IP expression [128]. This, in turn, enhances the radioresistance in recipient cells due to increased DSB repair kinetics [149].

A very similar protective effect, also mediated by exosomes, was observed in breast cancer (MCF7) cells under heat shock conditions [150]. Therefore, this phenomenon likely represents a broader stress response that is not confined to just radiation-induced stress, and serves to maximise the survival of the entire cell population to recurring stress signals [150]. However, a very recent study reported contradictory findings that show exosomal miRNA to be responsible for the radiosensitisation of recipient pancreatic cancer (MIAPaCa-2) cells [123]. In line with the previously stated role of miRNA species in transmitting RIBE via exosomes, miR-6823-5p was identified as a candidate that is responsible for increased ROS and DNA damage in recipient cells as a result of SOD1 inhibition [123]. Nevertheless, in contrast to the work discussed above [128,148], these phenotypes appeared to be enhanced when 5 Gy irradiation and radiation-derived exosome treatment were combined, as opposed to being mitigated [123]. Again, this highlights the complexity of RIBE and its associated influence on radiosensitivity.

While miRNA species clearly play a role in intercellular damage responses, the high diversity of cell lines, radiation types, and doses being used to study these effects have prevented a single model from being elucidated thus far. Nonetheless, the described exosome-mediated RIRE and radioprotective response represent cooperative cellular responses to radiation-induced stress, providing a population-wide mitigation of DNA damage upon recurring insult. While this would generally provide positive outcomes (such as enhanced survival) within normal tissues, the presence of these mechanisms within tumours may hinder treatment by radiotherapy. This could be mitigated by further pharmacological intervention, including PARP inhibitor combination therapy, warranting further investigation into the underlying basis of these mechanisms.

## 4. Conclusions

Non-coding RNA plays critical roles in both intracellular and intercellular DDR mechanisms. Within the cell, small and long non-coding RNA species generated both in trans (miRNA, snRNA and lncRNA) and cis (DDRNA, diRNA, dilncRNA, and nascent pre-mRNA) have distinct regulatory functions over DNA repair mechanisms. In particular, the DSB repair pathways HR and NHEJ are modulated by a combination of RNA-dependent mechanisms whose crosstalk can ultimately fine-tune these processes. These include controlling the expression or localisation of DNA repair proteins, the stabilisation of DNA ends for resection, and the scaffolding of multiple factors to promote repair. The current understanding of these mechanisms provides ample potential for pharmacological exploitation, exemplified by use of enoxacin to enhance DNA repair and cell survival [74]. Nevertheless, controversial findings have been discussed, which must be addressed to consolidate the opposing models of in cis transcription and DNA:RNA hybrid formation. In addition to cellular RNA levels, exosomal RNA content is influenced by radiation-induced damage. This leads to bystander effects in recipient cells, which can have positive or negative cellular outcomes. Canonical RIBE leads to DNA damage and cell death, whereas RIRE and the radioprotective response mitigate the damage caused by subsequent radiation insult. It remains to be seen whether these bystander effects are widespread or cell line-specific, as well as the exact interplay between the various mechanisms discussed here. However, their cellular context is important. The majority of the studies discussed here used cancer cell lines, particularly those of head and neck cancer, which is primarily treated using radiotherapy [148]. In this case, canonical RIBE within the tumour could lead to positive therapeutic outcomes while RIRE/radioprotective effects would be detrimental to the individual. Therefore, insight into the mechanisms of intercellular DDR could uncover novel treatment opportunities to overcome issues of radioresistance and tumour recurrence by enhancing RIBE or by alleviating RIRE/radioprotection. Nevertheless, the possibility of off-target effects in healthy cells should be carefully considered and highlights the need for further understanding of the mechanisms underlying RIBE and RIRE. In summary, RNA is a multifaceted DDR factor that can function in a variety of locations, and its study offers new insight into DDR and promise for therapeutic applications.

## Figures and Tables

**Figure 1 genes-12-01475-f001:**
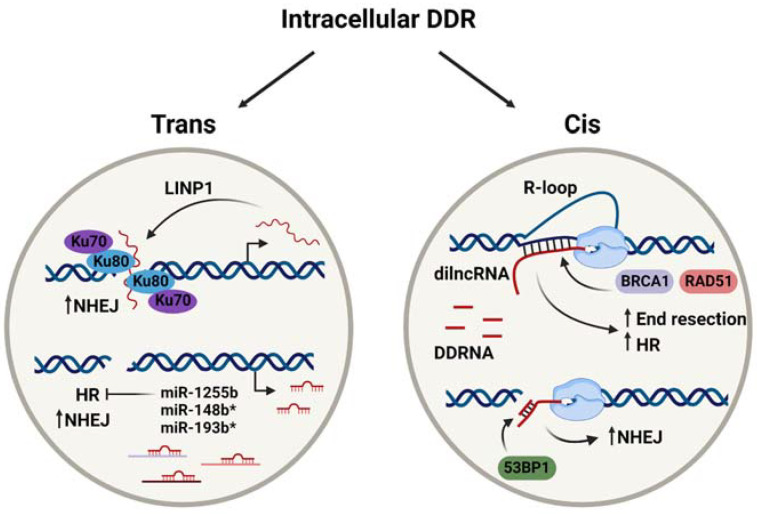
RNA contributions to intracellular mechanisms of DNA repair in cis and trans. Created with BioRender.com.

**Figure 2 genes-12-01475-f002:**
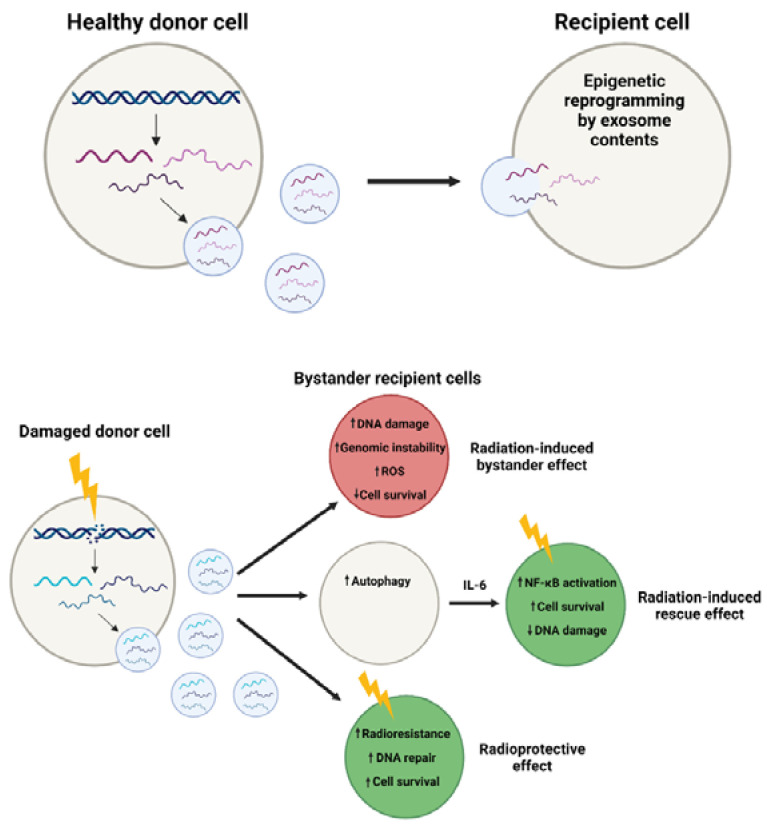
Radiation-derived exosomes can influence damage responses. Under normal conditions, RNA is packaged into exosomes and is taken up by recipient cells to allow intercellular communication. Irradiation of donor cells alters exosome contents. Uptake of these radiation-derived exosomes leads to various bystander effects in recipient cells. Created with BioRender.com.

**Table 1 genes-12-01475-t001:** lncRNA species and their roles in DDR.

lncRNA Species	Function	Reference
LINP1	Scaffolding of multiple Ku70-Ku80 dimers and DNA-PKcs at damage sites	[23,27]
NIHCOLE	Recruitment of Ku80 to damage sites and phase separation to drive NHEJ	[26]
LRIK	Binds damaged DNA, allowing DSB sensing and signalling to recruit Ku70	[24]
lnc-RI	Stabilises LIG4 and RAD51 expression by acting as a ceRNA for miR4727-5p and miR-193a-3p	[25,28]
HOTAIR	Promotes radioresistance via regulation of miR-454-3p, miR-218, and EZH2	[29,30,31]
HITT	Binds ATM, preventing MRN complex formation and restricting HR	[32]
LIRR1	Reduces expression of Ku70, Ku80, and RAD50	[33]
MEG3	Sequesters miR-182, enhancing radiosensitivity	[34]
BS-DRL1	Interacts with HMGB1 and facilitates its assembly on chromatin upon DNA damage in the brain	[35]
Aerrie	Associates with YBX1 and is recruited to damage sites to enhance DNA repair	[36]

**Table 2 genes-12-01475-t002:** miRNA species associated with DNA repair and the factors whose expression they alter.

miRNA Species	DNA Repair Factor(s)	Repair Pathway(s)	Reference
miR-138 miR-24	H2AX	HR, NHEJ	[49,50]
miR-100 miR-101 miR-421	ATM	HR	[46,51,52]
miR-1255b miR-148b * miR-193b *	BRCA1, BRCA2, RAD51	HR	[11]
miR-193a-3p miR-96-5p	RAD51	HR	[28,53]
miR-335 miR-130b	CtIP	HR	[54,55]
miR-101 miR-874-3p	DNA-PKcs	NHEJ	[46,53,56]
miR-502	Ku70, XLF	NHEJ	[57]
miR-622	Ku70, Ku80	NHEJ	[58]
miR-4727-5 pmir-1246	LIG4	NHEJ	[25,59]

Table adapted from Thapar, 2018 [4]. The * is part of the species name given in the reference.

**Table 3 genes-12-01475-t003:** RNA species with dysregulated expression in exosomes or bystander cells upon irradiation and their contribution to RIBE.

RNA Species	Gene(s) Regulated	Cellular Outcome	References
↑ miR-1246	*LIG4*	Decreased NHEJ efficiency	[59]
↑ miR-7-5p	*EGFR*	Increased autophagy	[117]
↑ miR-7	*BCL2*	Increased autophagy	[118]
↑ miR-663	*TGFB1*	RIBE suppression	[119]
↑ miR-34c	-	ROS induction	[120]
↑ miR-21	*SOD2, TGFB1*	ROS induction	[121,122]
↑ mir-6823-5p	*SOD1*	ROS induction	[123]
↑ mir-769-5p	*TGFBR1*	Reduced proliferation, increased oxidative damage	[124,125]
↑ mir-208a	*P21*	Increased proliferation, decreased apoptosis, increased radioresistance	[126]
↑ mir-96	*-*	Increased radioresistance	[127]
↑ miR-889	*DAB2IP*	Increased proliferation, radioresistance	[128]
↓ miR-365	*BCL2, CCND1, PIK3CA*	Increased proliferation, radioresistance	[110,128]
↓ miR-146a-5p ↓ miR-34a-5p	*N* *FKB1, BRCA1*	-	[110]

↑: upregulate; ↓: downregular.

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
