# Peer review of "Home and Away: The Role of Non-Coding RNA in Intracellular and Intercellular DNA Damage Response"

_genes, 2021, doi:10.3390/genes12101475_

Round 1

Reviewer 1 Report

A comprehensive and timely review on an ever more complex area of DDR. The review is well written, and a broad spectrum of references have been cited.

However, one thing I would alter is the Abstract. Much of the review focuses on double-strand break repair and this is not evident from reading the abstract.

Clearly the final formatting of the article needs to be improved if accepted for publication, as justification has led to some strange placements of the hyphen, including in the title. Other minor changes are listed below:

In text references need [number] after the authors name(s). Examples can be found on lines: 165/169/211/235/255/269/278/287/304/308/370/456/502/510/512/520

Line

35                        Start of the sentence seems out of order: Start with ‘NHEJ is initiated…..’ and add ‘end resection….’ later as appropriate.

41/233/295        Replace ‘which’ with ‘that’

78                        Delete ‘A handful’ and replace with something more scientific

148/151              Change ‘table 1’ to Table 1

156                      Figure 1 in the text: Should be Table 1.

187                      Define CTD

217                      Change ‘is’ to ‘are’

259                      Change ‘lead’ to ‘led’

282                      Add ‘RNA’ to polymerase (RNA polymerase)

334                      Figure 1 mentioned in text. Should be Table 1? If so, Table 1 only refers to miRNAs, so ‘Key examples’ at the start of the sentence is not correct, as that suggests more than is actually shown.

345/449              Figure 2 should be Table 2?

385                      Poorly expressed phrase, where does cisplatin fit in? Just needs rephrased, e.g., ‘…may lead to apoptosis in non-target cells in patients treated with cisplatin.’

402 etc                media or medium?! Please check

406/465/530     Table 2

447                      e.g. in italics

452                      Delete ‘cancerous cells’ and replace with ‘human cancer cell lines’

465                      Delete ‘a slight relief’ and replace with something more scientific

470                      Not Section 3.1.2 but 3.1.1, I think!

478                      Define PARP1

484                      vice versa in italics

Author Response

Thank you to this reviewer for their comments, we appreciate the feedback provided. Unfortunately there appears to have been an error during submission as the figures were uploaded as a separate document, but were not sent to reviewers with the manuscript. We apologise for any inconvenience caused by this. Figures have now been attached and we hope this clears up any confusion between figures and tables, both referenced in the text. The remaining comments have been addressed using "track changes" on the copy of the manuscript provided, including proper formatting of the document.

Reviewer 2 Report

This manuscript reviews the role of non-coding RNA in regulating the DNA damage response. First, the authors focus on the intracellular aspect, and how various non-coding RNAs (mainly lncRNAs, miRNAs, DDRNAs and dilncRNAs) mediate DNA repair and participate in repair pathway choice, more specifically after DNA double-strand breaks. In a second part, they discuss the role of RNA in the intercellular communication (through exosomes) between irradiated and unirradiated cells in their vicinity.

I have read this review with great interest, it is well-written and provides a good overview of the topic, as well as detailed examples. While a few other reviews about the role of RNA in DNA repair have been published recently, this work is original in combining intra- and intercellular aspects of the DDR mechanisms. The authors also discuss new and controversial data, which is useful for the field. I have some minor comments listed below.

- There is some confusion regarding the figures. In the text, the authors mention figures 1 and 2 as well as tables 1 and 2, while only tables 1 and 2 are present in the manuscript. After contacting the journal, I was told that figures 1 and 2 actually also refer to tables 1 and 2. In this case, the references to the figures in the text are not appropriate. For example, it is written lines 331-334 “Many species of non-coding RNA with direct contributions to repair of DNA lesions have been discussed here, includ-ing various lncRNAs and miRNAs, dilncRNA and DDRNA/diRNA. Key examples have been summarised in Figure 1.”, while table 1 is only about miRNAs involved in DDR and which DNA repair genes they regulate. Similarly for figure 2, lines 446-449 are introducing the radiation induced rescue effects and radioprotective effects, while the table 2 is about miRNAs involved in radiation-induced bystander effect.

The authors should consider adding graphical summaries of parts 2 and 3, this would make the review more attractive for the reader and easier to understand.

- An introduction about the effect of irradiation on cells is missing at the beginning of the 3rd part (which cellular pathways are activated following irradiation, what kind of DNA damage is generated and which DDR pathways deal with these insults, etc)

- Line 30: Double-strand breaks instead of double-stranded breaks

- Line 113: “ChIRP-qPCR”, maybe precise the abbreviation (Chromatin Isolation by RNA Purification)

- Line 140: through RNAi instead of though

- Table 1: please use the original paper as reference (Thapar, Molecules, 2018)

- Line 236: Maybe write a few words about what is enoxacin/ what this compound was already known to do before this study

- Line 238: “DNA:RNA hybrids” instead of DNA hybrids?

- Line 292: Complementary rather than opposing?

- Lines 304, 310 and 311: Maybe replace “genic” and “non-genic” by “intragenic” and “intergenic”

- Line 349: “remain elusive while widely studied”

- Lines 382-386: Please clarify the link between miR-3168 content in IR-treated cells/exosomes and cisplatin-induced apoptosis in recipient cells.

- Lines 398-401: I do not understand the meaning of “specific dependence of this phenomenon on exosome-packaged proteins has not yet been reported.”, while it is said above that RNA and protein work synergistically.

- Lines 542-549: I would suggest describing this example in part 3.1.2

Author Response

Thank you to this reviewer for their comments, we appreciate the feedback provided. Unfortunately there appears to have been an error during submission as the figures were uploaded as a separate document, but were not sent to reviewers with the manuscript. We apologise for any inconvenience caused by this. Figures have now been attached and we hope this clears up any confusion between figures and tables, both referenced in the text. Specific comments have been addressed using the "track changes" function on the copy of the manuscript provided, including the addition of a small introduction to the effect of irradiation on cells in part 3 and further details about enoxacin. 

Reviewer 3 Report

The manuscript ‘Home and away: The role of RNA in intracellular and intercellular DNA Damage Response’ by Shaw A. and Gullerova M. is a well written and timely review discussing the role of RNA in mediating the DNA Damage Response (DDR) especially during double-stranded DNA breaks (DSB). Although it was presumed that proteins are the only factors involved in DSB, recent years have seen an explosion of evidence placing RNA as a key player in DDR. The authors first discuss intracellular RNAs that mediate DDR such as long non-coding RNAs, miRNA and DDRNAs. They judiciously distinguish between RNA acting in trans and those acting in cis to the DNA break site. They then describe the role of intercellular RNAs in promoting radiation induced bystander effect (RIBE) or radiation-induced rescue effect (RIRE), especially relevant during radiotherapies, focusing on miRNAs. Although clear mechanistic understandings of RIBE and RIRE are lacking, the authors nicely illustrate the current knowledge on miRNA-induced RIBE or RIRE and emphasize the difficulty of comparison due to the use of different radiation types, doses etc between studies. Altogether, this review discusses the emerging theme of RNA in DNA damage and will be of broad interest to the RNA and DNA repair communities.

Major comments:

  • Although the authors briefly mentioned that RNA can also trigger DNA damage in section 2.2.2 and 3.1.2, the authors should clearly indicate in the Introduction the notion that RNA can induce DNA damage and is not just involved in DDR. For instance, R-loops and their dysregulation have been extensively linked to DNA damage impacting genomic integrity (PMID: 22541554, 25233079 etc).
  • In section 2.1.1, the reader is left with the impression that lncRNAs only act as scaffolds to recruit factors promoting DDR and wonders if these lncRNAs form DNA-RNA hybrid structures at the DNA lesion sites. The authors should comment if such structures are formed or if there is some complementarity between parts of the lncRNAs and the region of the DSB.
  • The authors missed the opportunity to discuss another emeging class of RNA involved in DDR that is snRNA incorporated in snRNPs (PMID: 27462460, 27991914). This class of non-coding RNAs should be added to section 2.1.

Minor comments:

  • Line 345- 346: Figure 2 is mentioned but not present in the manuscript.
  • It would be helpful to have a table listing the lncRNAs involved in DDR similar to table 1 for the miRNA.
  • Line 105 it is unclear what the authors mean by ‘hyper stoichiometry’. Although this term is used in chemistry, it is unclear if it can be used for biological systems.

Author Response

Thank you to this reviewer for their comments, we appreciate the feedback provided. Unfortunately there appears to have been an error during submission as the figures were uploaded as a separate document, but were not sent to reviewers with the manuscript. We apologise for any inconvenience caused by this. Figures have now been attached and we hope this clears up any confusion between figures and tables, both referenced in the text. Specific comments have been addressed using the "track changes" function on the copy of the manuscript provided. In particular, the introduction has been amended to include the role of RNA and R-loops in DNA damage induction and a sub-section has been added to section 2.1 detailing the contribution of snRNA to DDR. An additional table summarising more examples of lncRNAs involved in DDR has been included, which demonstrates lncRNA contribute more than scaffolding functions. 

Reviewer 4 Report

The manuscript by Shaw and Gullerova aims to review our current knowledge on the roles of RNA molecules in the DNA damage response (DDR). While this emerging and active research topic has already been the focus of a large number of reviews, the authors put forward here the underappreciated contribution of RNA in intercellular communication in the DDR. However, in its present from, the text solely summarizes published findings, failing to untie their complexity or to reconciliate their apparent discrepancies. These issues should be addressed so that this manuscript can actually provide a valuable addition to the existing literature.

  1. In view of the diversity of molecular mechanisms that are detailed in the text, the manuscript suffers from the absence of figures. The text refers to Figures 1 and 2, which are absent from the submitted version, and it is unclear whether this solely relates to Tables 1 and 2. Appropriate illustrations need to be included in order to guide the readers in their understanding of the described processes.

  1. In general, the authors detail the findings from the literature but fail to provide the big picture. This is particularly striking throughout sections 2.2 and 3. I would suggest that the authors revise the concluding paragraphs in these sections to convey clear and balanced take-home messages.

  1. There are parts of the literature related to the manuscript topic that are unmentioned, or not properly covered.

- There is a number of reports on the role of RNAs as templates in DSB repair (e.g. doi:10.1016/j.molcel.2020.08.011 and references therein). This concept is currently absent in the manuscript and should be described.

- Beyond the scaffold role of characterized lncRNAs in the activity of NHEJ factors, nascent pre-mRNAs have also been scored in association with the NHEJ machinery (doi:10.1038/ncomms13049). This study should be mentioned.

- The authors should distinguish the situations in which RNAs play a role in the DNA damage response from those where RNAs are themselves genotoxic intermediates that further trigger DNA repair. In section 2.2.2, the authors report the evidence that DNA:RNA hybrids or R-loops formed upon DSB induction function in HR (e.g. references #74 and #75). This is distinct from the case of the referenced S. cerevisiae study (reference #71) where hybrids are the source, rather than the consequences, of the accumulation of DNA damages (likely DSBs, which are further fixed through canonical HR mechanisms).

- In line with the variety of RNA and DNA:RNA hybrid species generated following DSB induction, the authors should also mention the recent report that ssRNAs participate in RAD52-dependent repair mechanisms (doi:10.1038/s41586-020-03150-2)

Author Response

Thank you to this reviewer for their comments, we appreciate the feedback provided. Unfortunately there appears to have been an error during submission as the figures were uploaded as a separate document, but were not sent to reviewers with the manuscript. We apologise for any inconvenience caused by this. Figures have now been attached and we hope this clears up any confusion between figures and tables, both referenced in the text. Specific comments have been addressed and will be clear from the "track changes" tab in the copy of the manuscript provided. In particular, the concluding paragraphs for each section have been amended to better summarise the big picture and take home message of that particular section. The suggested studies have also been included.

Round 2

Reviewer 4 Report

The authors have now addressed my previous concerns. The figures which are visible in this revision are a good addition to the text.

Author Response

We are glad that this reviewer finds the figures useful.